# Glucocorticoid and PD-1 Cross-Talk: Does the Immune System Become Confused?

**DOI:** 10.3390/cells10092333

**Published:** 2021-09-06

**Authors:** Sabrina Adorisio, Lorenza Cannarile, Domenico V. Delfino, Emira Ayroldi

**Affiliations:** Department of Medicine and Surgery, Section of Pharmacology, University of Perugia, 06100 Perugia, Italy; adorisiosabrina@libero.it (S.A.); lorenza.cannarile@unipg.it (L.C.); domenico.delfino@unipg.it (D.V.D.)

**Keywords:** PD-1/PD-L1, glucocorticoids, cancer, immune response

## Abstract

Programmed cell death protein 1 (PD-1) and its ligands, PD-L1/2, control T cell activation and tolerance. While PD-1 expression is induced upon T cell receptor (TCR) activation or cytokine signaling, PD-L1 is expressed on B cells, antigen presenting cells, and on non-immune tissues, including cancer cells. Importantly, PD-L1 binding inhibits T cell activation. Therefore, the modulation of PD-1/PD-L1 expression on immune cells, both circulating or in a tumor microenvironment and/or on the tumor cell surface, is one mechanism of cancer immune evasion. Therapies that target PD-1/PD-L1, blocking the T cell-cancer cell interaction, have been successful in patients with various types of cancer. Glucocorticoids (GCs) are often administered to manage the side effects of chemo- or immuno-therapy, exerting a wide range of immunosuppressive and anti-inflammatory effects. However, GCs may also have tumor-promoting effects, interfering with therapy. In this review, we examine GC signaling and how it intersects with PD-1/PD-L1 pathways, including a discussion on the potential for GC- and PD-1/PD-L1-targeted therapies to “confuse” the immune system, leading to a cancer cell advantage that counteracts anti-cancer immunotherapy. Therefore, combination therapies should be utilized with an awareness of the potential for opposing effects on the immune system.

## 1. Introduction

Inhibitory receptors, such as programmed cell death 1 receptor (PD-1), are involved in immune response regulation, including the response against tumors. Targeting and blocking PD-1 and its ligands, programmed cell death receptors ligand 1/2 (PD-L1/2), using monoclonal antibodies (mAb), is currently one of the most effective immunotherapy approaches for some cancers. Inhibition of the PD-1/PD-L1 system and, more generally, the immune checkpoint blockade has transformed and improved the management of neoplastic pathologies [1].

Upon ligand binding, PD-1 inhibits T cell receptor (TCR) signaling resulting in decreased T cell effector function, T cell proliferation, and immunostimulatory lymphokine production. This pathway is one of the mechanisms by which autoimmunity is prevented, contributing to immune tolerance maintenance [2]. The inhibition of T cell-mediated immunity is also a strategy that cancer cells use to escape immune surveillance. Cancer cells have been found to upregulate inhibitory checkpoint molecules, resulting in suppression of cellular and humoral immunity [1,3]. Therefore, targeting and blocking PD-1/PD-L1, enhances T cell-mediated immunity against these cancer cells. In particular, targeting memory, tumor-infiltrating PD-1 + T cells plays a major role in the efficacy of this treatment strategy [3].

Endogenous glucocorticoids (GCs) are controlled by the hypothalamic–pituitary–adrenal (HPA) axis. GC secretion is induced in response to external and internal stress stimuli. Dysregulation of the HPA axis can result in increased GC secretion, which has been associated with stress and aging related cancers [4,5,6]. Exogenous GCs are currently used as therapy for inflammatory and autoimmune diseases, as well as certain myelodysplastic syndromes [7]. For solid tumors, GCs are often used as adjuvant therapy, although their effects on the development and spread of these tumors are debated [8].

The immune system is the primary target for GC treatment. GCs have been shown to inhibit T cell activation, interfere with TCR-induced gene expression, and inhibit dendritic cell maturation [9,10]. This suggests that alterations in GC secretion during chronic stress or exogenous long-term GC administration results in immunosuppression and a sub-optimal immune response against cancer cells. GCs affect both systemic and tumor microenvironment immune cells, leading to alterations in immune signaling and neoplastic cell biology [11]. The concomitant and often overlapping effects of GCs may also involve the PD-1/PD-L1 system. Often, immunotherapy-related adverse reactions require high-dose GC administration [12], which may interfere with the efficacy of immunotherapy.

In oncology, the use of PD-1/PD-L1 inhibitors is a relatively recent therapeutic modality, while the use of GCs as adjuvants to chemotherapy is long-established. Importantly, the effects of the combination of these two therapies are not well understood. Here, we discuss combined GC and PD-1/PD-L1 inhibitor therapy and if this combination is necessary or rational in all cases, with a focus on the effects of these drugs on the immune system.

## 2. PD-1/PD-L1 in Health and Cancer

### 2.1. PD-1/PD-L1 Function

The immune checkpoints, PD-1 and cytotoxic T-lymphocyte–associated antigen 4 (CTLA-4), negatively regulate T cell immune function. In concert with regulatory T cell (Tregs), these molecules contribute to the maintenance of peripheral tolerance, preventing activated T cells from excessive proliferation and effector function. PD-1 is widely expressed in immune cells. It is expressed at low basal levels on resting naïve T cells and in some populations of thymocytes. During an immune response, PD-1 is transiently expressed on CD4+ and CD8+ T cells, B cells, natural killer (NK) cells, monocytes, dendritic cells (DCs), and tumor-infiltrating lymphocytes (TILs) [13,14]. TCR engagement and cytokine stimulation, including interleukin (IL)-2, IL-7, IL-15, IL-21, tumor necrosis factor-α (TNFα) and interferon gamma (IFNγ), induce PD-1 expression following antigenic stimulation, which then decreases when the antigen stimulus stops [15,16,17]. PD-1 expression is also modulated by glycolysis, which downregulates PD-1 in CD4+ T cells. In macrophages, PD-1 is induced by lymphokines (IL-6, IL-1β, IFNγ, TNFα) and Toll-like receptor ligands, including lipopolysaccharide (LPS) [18,19]. In B cells, LPS, IL-4, and IFNγ downregulate the B-cell receptor-induced expression of PD-1 [20]. In addition, chronic inflammation and cancer upregulate PD-1 levels in immune cells [21]. PD-1 is upregulated on CD8+ and CD4+ T cells during pathogen- or tumor-induced activation, serving to limit the activity of T cells [21].

The over-stimulation of T cells due to a deficiency in PD-1 expression and function can be detrimental. PD-1-knock-out (KO) mice develop autoimmune dilated cardiomyopathy [22], followed by a lethal multiorgan autoimmune syndrome [2]. Polymorphisms of programmed cell death 1 (PDCD-1) gene are associated with autoimmune diseases, cancers, and viral infections [23]. PD-1-KO cytotoxic T lymphocytes (CTLs) [24] and PD-1-deficient CD8+ transgenic T cells [25] have been shown to have enhanced anti-tumor activity in vivo.

In humans and mice, PDCD1 contains five exons. Exon 1 encodes a signal sequence, exon 2 an IgV-like domain, exon 3 a stalk and transmembrane (TM) domain, and exons 4 and 5 the cytoplasmic domain that has an immune-receptor tyrosine-based inhibitory motif (ITIM) and an immunoreceptor tyrosine-based switch motif (ITSM) [26]. Four splicing variants have been identified whose functions have yet to be determined. However, the splice variant that lacks exon 3 codes for a soluble PD-1 (sPD1) that does not bind PD-L1 and PD-L2 but has been detected in inflammatory and autoimmune diseases. Thus, this form could be a potential biomarker for disease [27,28].

The molecular mechanisms responsible for PD-1 expression have been studied primarily in CD8+ T cells. Briefly, nuclear factor-activated T cells c1 (NFATc1) is necessary for the initial TCR-induced PD-1 expression [29]. However, other transcription factors also contribute to PD-1 expression, including the signal transducers and activators of transcription (STATs) that are activated by lymphokines, activating protein-1 (AP-1), and interferon regulatory factor 9 (IRF9). In the later phase of CD8+ activation, PD-1 expression is downregulated by B lymphocyte-induced maturation protein 1 (Blimp-1), which represses PD-1 directly by blocking NFATc1 expression [30]. Further, T-bet directly represses the transcription of PD-1 during chronic infection [31] (Figure 1A).

The two known PD-1 ligands—PD-L1, also known as B7 homolog 1 (B7-H1); and PD-L2—bind to PD-1 with different affinities, the latter having a higher binding affinity [32]. PD-L1 and PD-L2 are type 1 glycoproteins, with IgV and IgC domains. PD-L1 is expressed on antigen presenting cells (APCs), including DCs and macrophages, tumor cells, T cells, B cells, and non-lymphoid tissues (pancreatic islet cells, cardiac endothelium). PD-L1 is also expressed in sites of immune privilege, such as the placenta, testes, and eye [33]. In contrast, PD-L2 expression is induced upon macrophage and DC activation [34]. PD-L1 contains seven exons that encode the 5′UTR, the signal sequence, IgV-like domain, IgC-like domain, transmembrane and intracellular domains, part of the intracellular domain, and the 3′UTR [35].

PD-L1 transcription is regulated by multiple pathways and occurs independently of PD-1 transcription [36]. Persistent pathogen infections lead to pro-inflammatory cytokine production that induce PD-L1 upregulation. In addition, IL-10, IL-2 TNFα, and IFNs may induce PD-L1 expression [37], while IL-4, granulocyte-macrophage colony-stimulating factor (GM-CSF), and IFNs induce PD-L2 expression [38,39]. Hypoxia-inducible factor-1α (HIF-1) and STAT-3 act on the promoter of PD-L1. The main transcription pathway for IFN-γ-induced PD-L1 expression is through JAK/STAT [40], whose activity is mediated by interferon-responsive factors (IRFs) [41] and the nuclear factor-κB (NF-κB) pathway. Likewise, TNF-α upregulates PD-L1 expression via the TNF-α/NF-κB pathway [42]. In tumor cells, PD-L1 is upregulated not only through tumor-promoting pathways, such as mitogen-activated protein kinase (MAPK) and phosphatidylinositol 3-kinase (PI3K)/protein kinase B (Akt), but also by factors from the tumor immune microenvironment (TIM) and by a hypoxic microenvironment, which stimulates HIF-1 expression [39] (Figure 1B).

PD-L1/2 on APCs or tumor cells binds PD-1 on T cells and initiates an intracellular signaling cascade leading to the impairment of TCR signaling and lymphokine production. PD-1 engagement results in the phosphorylation of its ITSM/ITIM motifs, recruitment of Src homology region 2 domain-containing phosphatase (SHP)1/2, dephosphorylation of proximal TCR components, and inhibition of CD28-mediated activation of PI3K. This sequence results in reduced T cell proliferation, decreased immunostimulatory lymphokine production, including IL-2, TNF, and IFN-γ, reduced T cell killing ability, and diminished T cell transcription factor activity, such as GATA-3, T-bet, NF-κB, Ap-1, and nuclear factor of activated T cells (NF-AT) [23,39,43]. PD-1 activation inhibits two important signaling pathways, PI3K-Akt and Ras-Mek-Erk, most likely via SHP-2 dephosphorylation of phospholipase C gamma 1 (PLCγ1) [43]. Inhibition of PI3K prevents the expression of B-cell lymphoma-extra-large (BCL-xL), while the inhibition of Ras-Mek-Erk increases the expression of the proapoptotic BIM, thus promoting apoptosis. In addition, the inhibition of Ras-Mek-Erk blocks downstream cyclin-dependent kinase-2 (CDK2) activation, halting cell cycle progression [43].

In addition to its effects on T cell function and proliferation, PD-1 also inhibits the development of T helper type 1 (Th1) and Th17 cells and promotes Treg differentiation [44]. PD-1 inhibits glycolysis, required for the differentiation of naïve T cells into the Th1 and Th-17 lineages. The activation of this receptor also upregulates phosphatase and TENsin homolog deleted on chromosome 10 (PTEN), which, together with the downregulation of phospho-Akt, mammalian target of rapamycin (mTOR), and extracellular signal-regulated kinase 2 (ERK2), is responsible for the increase in Treg activity. Tregs contribute to the inhibition of T cell activation and proliferation [44] (Figure 2). This negative feedback induces anergy, particularly in effector cytotoxic CD8+ T cells, the cells responsible for the anti-tumor immune response. Tissue expression of PD-L1 and PD-L2 is therefore protective for local tissue integrity by controlling the T cell response, which, if excessive, could lead to tissue inflammation and damage.

### 2.2. PD-1/PD-L1 and Cancer

Immune checkpoints help shape the dynamic cell signaling networks between tumor cells and the tumor immune microenvironment (TIME), crucial for either tumor cell elimination or survival. An important indicator of cancer proliferation, metastases, and recurrence is the presence of cancer stem cells (CSCs), which have been identified in many cancers [45]. These cells replicate, differentiate, and induce tumor chemoresistance. CSCs also have immunosuppressive functions through interactions with the TIME and protect against immune cell attack [46]. T cells that are stimulated by tumor antigens produce cytokines that help induce an adaptive immune response. This process is controlled by positive and negative immunological stimuli, resulting in a critical balance that ensures the efficiency of the immune response. Tumor cells exploit and amplify negative stimuli for self-protection, including modulating the PD-1/PD-L1 system.

PD-L1 is upregulated in many tumors and tumor stromal cells, such as tumor-associated macrophages and myeloid-derived suppressor cells, that negatively regulate the adaptive and innate immune responses to cancer [47]. PD-L1 upregulation, observed in some tumors, is induced by pro-inflammatory cytokines, such as IFN-γ and TNF-α [47]. During an anti-tumor immune response, PD-1 expression in TIME lymphocytes is higher than that of peripheral lymphocytes [48]. This leads to an increase in the function of the PD-1/PD-L1 system in the TIME, with down-modulation of the TCR signalosome in tumor-infiltrating CD8+ T cells, leading to impairment of the immune response to the tumor [48,49]. In TILs, the enhanced PD-1/PD-L1 interaction induces cell cycle arrest at the G0-G1 phase. This arrest occurs through CDK inhibition as well as TCR internalization [50]. PD-1 engagement stimulates transcription of the E3 ubiquitin ligases, of the CBL family, which ubiquitinate the TCR chains, leading to down-modulation of the TCR, thereby decreasing its antigenic response capacity [48,50,51].

In addition to the pathway described above, a direct correlation has been demonstrated between PD-L1 expression and stemness-associated genes [52,53,54]. For example, one study found an association between PD-L1 and breast cancer stemness score in breast cancer samples [52]. PD-L1-silencing induced a decrease in the expression of embryonic stem cell transcriptional factors, such as OCT-4A, Nanog, and the stemness factor BMI, through the PI3K/Akt pathway [52]. PD-L1, through activation of PI3K/Akt in CSCs, induces the expression of OCT-4 and Nanog, important for maintaining pluripotency, and drives stemness and proliferation of cancer cells by regulating BMI and mTOR expression [53,54]. These observations suggest an active, transductive role for PD-L1.

In examining PD-1/PD-L1’s crucial role in cancer cells and CSC survival, researchers have focused their efforts over the last 15 years on targeting and blocking this inhibitory pathway. Studies have focused on the use of monoclonal antibodies (mAbs) that bind to PD-1 or PD-L1, preventing the PD-1/PD-L1 interaction, resulting in the restoration of the anti-cancer immune response. Recognition of tumor antigens by a committed CD8+ T cell is necessary for an effective pharmacological target of the PD-1/PD-L1 system to be functional. Blocking PD-1/PD-L1 through mAb binding activates tumor-specific CD8+ T cells by allowing the activation of the TCR and CD28 co-stimulation with DCs, both in the TIME and the tumor-draining lymph node (tdLNs) [55]. However, clinical data have shown that PD-L1-targeted immunotherapy is effective in patients who do not express PD-L1 in the TIME [56]. Indeed, many authors suggest that PD-L1 blockade may also facilitate the de novo priming of tumor-specific CTLs in tdLNs [57].

Many mAbs are now available for cancer therapy use and have been approved for the treatment of various cancers, improving the survival and quality of life for patients. However, a negative effect of this treatment is that the PD-1/PD-L1 blockade may induce autoreactive T cells responsible for immune-related adverse effects (irAEs) [58]. In addition, recent evidence suggests that one of the molecular mechanisms responsible for the failure of PD-1/PD-L1 targeted therapy is the presence of extra-tumoral PD-L1 contained in exosomes, small nanoparticles released from the surface of both normal and tumor cells containing biologically active molecules [59]. Exosomes may release pro-tumoral signaling molecules in cancer. Preclinical and clinical evidence has associated the presence of PD-L1-containing exosomes with an immunosuppressive TIME and with a worse prognosis in several cancers [59]. In addition, PD-L1-containing exosomes could contribute to PD-1-specific mAb treatment resistance. PD-L1 expression on the exosome surface could allow the tumor to exert an immunosuppressive effect, not only locally but also at a distance, wherever the exosomes are found [60]. PD-L1 expression on tumor-derived exosomes in the peripheral blood of patients positively correlated with the spread and stage of disease [60]. Accordingly, it has been suggested that removal of exosomes containing PD-L1 could enhance the effectiveness of anti-PD-1/PD-L1 immunotherapy [61].

## 3. Glucocorticoids in Health and Cancer

### 3.1. Immune Functions of Glucocorticoids

GCs are steroid hormones that control numerous physiological processes, such as cell growth and differentiation, morphogenesis, metabolism, electrolyte and water balance, inflammation, cognitive functions, cardiovascular function, and immune responses [62,63]. These actions regulate the body’s adaptive response to stress. GCs are produced and secreted by the adrenal cortex and their production is regulated by the HPA axis [64]. At pharmacological concentrations, GCs display potent anti-inflammatory and immunosuppressive effects, resulting in modulation of both the innate and adaptive immune response [65]. Therefore, GCs have become the most widely used anti-inflammatory and immunosuppressive drugs worldwide [7,66]. GCs modulate immune system function through the glucocorticoid receptor (GR), a transcription factor that regulates the expression of various receptors, cytokines, adhesion molecules, and other factors crucial for immune system activity [67].

Both genomic and non-genomic effects of GCs involve the modulation of numerous inflammatory signaling pathways and transcription factors, including AP-1, NF-κB, cAMP response element-binding protein, GATA-3, GATA-1, IRF3, and NF-AT. GCs have been shown to inhibit the activity of NK cells, cyclooxygenase-2 (COX-2), inducible nitric oxide synthases (iNOS), and lymphokine expression in neutrophils. In addition, GCs promote a tolerogenic phenotype in macrophages and DCs, decreasing co-stimulatory molecule expression and chemokine production [67]. GCs inhibit T cell activation by decreasing activation of key transcription factors, such as NF-κB, AP-1, and NF-AT, and the expression of kinases involved in TCR signaling [67]. GCs also inhibit T cell activation through rapid, non-genomic effects. These effects involve reducing early phosphorylation events after TCR ligation [68] and the rapid destruction of multiprotein complexes containing GR, lymphocyte-specific protein tyrosine kinase (LCK), and proto-oncogene tyrosine-protein kinase (FYN) associated with the TCR, resulting in decreased signaling downstream of the TCR [69]. An increase in Treg development, modulation of naïve CD4+ differentiation (inhibition of Th-1- and Th-17-cells and promotion of Th-2 cells), and impairment of B-cell function result from GC-mediated immunosuppression (Figure 3). Immunosuppression is the desired effect for anti-inflammatory and immunosuppressive drugs, but it becomes a negative side effect during GC adjuvant cancer therapy.

### 3.2. Glucocorticoids and Cancer

While the pharmacological administration of GCs is used for the treatment of many hematologic malignancies by exploiting their anti-apoptotic capability, their role in solid cancers is less clearly defined. GC therapy for solid tumors is often used to control chemotherapy and immunotherapy side effects. However, GCs may promote either tumor suppression or tumor progression, depending on cancer type and the phase of neoplastic disease and the immune system efficiency [70]. GCs’ effects on solid tumor development or inhibition may be indirectly linked to their effects on the immune system but may also be due to direct changes induced by GR activity in the neoplastic cell itself [71] (Figure 4).

As previously stated, endogenous and exogenous GCs lead to immunosuppression, resulting in an ineffective immune response against various cancers [72,73,74]. This effect involves all cells of the immune system (Figure 3) in the TIME but also systemically. The molecular mechanisms and pathways leading to GC-dependent immunosuppression appear to be cancer-type dependent, explaining the difference in the effects of GCs on different tumors [71]. For example, in breast cancer, GCs are responsible for an increased tumorigenic TIME through induction of chronic inflammatory pathways and suppression of adaptive inflammatory responses [75]. Excess GCs, observed in adrenocortical carcinomas, worsen disease prognosis through the depletion of TILs [76]. Tumor immune evasion can be promoted by the tumor itself, through GC autocrine secretion, as seen in human skin squamous cell carcinoma and melanoma colon-cancer cells [77,78].

However, GCs also directly influence the biological functions of the neoplastic cell. Cancer cells, like normal cells, respond to GCs according to the number and type of GR isoforms expressed and GR affinities for specific GCs [79,80,81]. Studies have associated GR genetic changes to the development of various tumors. In vivo carcinogenesis models demonstrate that GC-loss protects mice against the development of tumors [82]. However, GR expression and function is specific for each type of neoplastic cell, leading to different results for each neoplasm [71]. For example, in ovarian and endometrial cancer cells, increased GR expression has been associated with poor prognoses [83,84]. GR expression is also correlated with a poor prognosis in estrogen receptor alpha (ERα)-negative breast cancer; however, it is correlated with a good prognosis in ERα-positive breast cancer [85]. An increase in stress-driven GCs has been associated with breast cancer progression [85,86]. In contrast, an in vitro study using cancerous epithelial prostrate cells observed a favorable prognosis for non-small cell lung cancers expressing GR, suggesting that GR may function as a tumor suppressor in these contexts [87,88].

GCs stimulate either pro-proliferative or inhibitory pathways in cancer cells. For example, GCs can induce activation of pro-proliferative p38-MAPK and Akt signaling pathways or can inhibit the tumor suppressor protein p53 [4,89]. GCs can also inhibit the synthesis and release of prostaglandins in prostate cancer cells, slowing prostate cancer growth and spread, while controlling pain [89]. GCs have been shown to inhibit the growth of patient-derived pancreatic cancer cells using a xenograft model, through the suppression of NF-kB, epithelial-mesenchymal transition (EMT), IL-6, and vascular endothelial growth factor (VEGF) [90]. On the other hand, GCs can induce resistance to apoptosis in epithelial tumor cells through various mechanisms, including upregulation of IκBα, serine/threonine survival kinase 1 (SGK1), and mitogen-activated protein kinase phosphatase (MKP1/DUSP1) [89,91].

## 4. GC and PD-1/PD-L1 Pathways in Health and Cancer

### 4.1. Intersection of GC and PD-1/PD-L1 Pathways

Endogenous GC and PD-1/PD-L1 signaling pathways, in healthy conditions, work together to continuously monitor and balance the immune response. TCR signaling upregulates the expression of PD-1 on the T cell surface. IFN-γ, released from activated CD8+ T cells, upregulates PD-L1 expression on APCs and stromal cells. PD-L1 then binds to PD-1 and triggers regulatory effects [13,92]. The PD-1/PD-L1 pathway exerts a “tonic” control over the T response, sharing this role with other membrane inhibitory receptors and with hormonal stimuli, such as GCs. The balance between immunosuppressive and immune-stimulating pathways orchestrates a well-organized and efficient immune response [13].

In addition to their physiological functions, GC and PD-1/PD-L1 pathways may also play a role in the pathogenesis or control of numerous diseases. Dysregulation, including the increase or inhibition of signaling, can give rise to diseases such as autoimmune and neoplastic disorders. It has recently been shown that GCs upregulate PD-1 expression on T and NK cells [93,94]. This suggests that (1) GC potentiation of the inhibitory effects of PD-1/PD-L1 signaling may be useful in therapeutic strategies for autoimmune diseases; (2) using GCs as adjuvant treatment during cancer therapy may lead to therapeutic failure in some tumors [95]; and (3) GC-induced expression of PD-1 on NK cells may lead to protection against cytokine-mediated disease during viral infection [96,97]. These claims have been demonstrated in both preclinical and clinical trials studies using GC- and PD-1/PD-L1-blocking drugs.

The GC-induced upregulation of PD-1 on T cells [95], especially on TILs, results in an increase in the PD-1/PD-L1 inhibitory capacity. This GC-mediated genomic effect is dependent on GR-induced transcription of PD-1, which contributes to the anti-inflammatory and immunosuppressive activity of pharmacological doses of GCs [93]. This mechanism contributes to the efficacy of GCs in the treatment of autoimmune disease but may also be responsible for tumor immune evasion [95]. In addition, GCs cooperate with cytokines, detected at the tumor site, to induce PD-1 expression on human NK cells (CD56brigh tumor-infiltrating NK cells) [94]. This is mediated directly through PD-1 transcription and also through translational regulation.

Upregulation of PD-1 contributes to immunotherapy resistance by reversing tumor growth inhibition. GC-induced PD-1 upregulation on NK cells may be a good candidate pathway for immunotherapy, through pharmacological blocking, for example, of PD-1 transcription or translation [94,96]. However, caution is warranted given that PD-1 expression on NK cells, induced by endogenous GCs, protects against cytokine-mediated inflammation during viral infection [97].

### 4.2. Cross-Talk between GC and PD-1 Pathways in Tumor Immune Evasion

T cells, stimulated by tumor antigens, migrate to the tumor site and together with other cells, including NK cells, constitute the TIME that determines the tumor fate. An immunosuppressive TIME promotes tumor growth and spread [98]. Although their physiological roles and molecular mechanisms in regulating the immune system are different, PD-1/PD-L1 and GCs have similar effects, particularly on T cells. Both inhibit the activity of T cells, including TILs, and other cells of the immune system, thus contributing to tumor immune evasion. Increased expression of PD-1 on T cells results in a dampening of T cell activation, thereby avoiding an exaggerated response that could be harmful if directed towards self-antigens. Therefore, any stimulus that induces PD-1 expression on activated T cells protects cancer cells from immune attack [33].

Indeed, blocking PD-1/PD-L1 has become an important therapeutic approach for several cancers, aimed at enhancing T cell-mediated anti-tumor immunity. The oldest example of a mAb against PD-1 is nivolumab, first used for metastatic melanoma [99] and now demonstrated to be effective in an increasing number of cancers [33,47,100]. However, anti-PD-1 antibodies may activate irAEs, affecting lung, gastrointestinal tract, skin, liver, the endocrine system, and other organ systems.

#### 4.2.1. Role of Exogenous GCs in PD-1/PDL-1 Blockage

GC use for the management of immunotherapy side effects involves both basic and clinical immunology and, based on conflicting scientific observations, there are differing opinions on therapeutic modality. As mentioned, GC-induced PD-1 expression on T and NK cells may contribute to tumor immune escape, which impedes cancer therapies [93,94]. In fact, the depletion of TILs induced by GCs in adrenocortical carcinoma is responsible for resistance to immunotherapy [76]. However, researchers have demonstrated that in mice transplanted with cancer cells and treated with dexamethasone (DEX), a synthetic GC, in doses comparable to those received by patients with cancer, DEX induces lymphodepletion of peripheral lymphocytes but does not affect the expression of immune checkpoint molecules, including PD-1 on TILs [101]. Based on this data, the authors argue that GCs may have deleterious effects on systemic activity only and that their use is safe when used in combination with immunotherapies [101].

The numerous clinical observations regarding GC usage for controlling immune therapy side effects lead to different conclusions. Systemic use of methylprednisolone may control most irAE [12,102]. For example, rheumatological disease, such as rheumatic polymyalgia or synovitis, has been associated with PD-1-inhibitor therapy in patients with kidney carcinoma. The response of these patients to oral GCs was satisfactory and GCs did not affect the efficacy of immunotherapy [103]. GCs, used systemically for the therapy of irAEs, do not modify overall survival. In contrast, if GCs are used for other complications due to PD-1/PDL-1 blocking therapy, they can impair its efficacy. Inhaled and topic GCs have no effect on overall survival [104]. There are many other examples of cancer immunotherapy side effects being effectively treated with GCs [105]. However, due to the complexity of immunotherapy effects, the large number of complications, and the possibility that GC-induced immunosuppression could counteract the activity of PD-1/PD-L1-blocking drugs, combining the two therapies, when not mandatory, is still problematic. Definitive guidelines on GC treatment concurrent with PD-1/PD-L1 pathway inhibitors are still lacking [106]. Each neoplasm, given the side effects of immunotherapy, would need a specific protocol for dosage and duration of GC administration.

#### 4.2.2. Role of Endogenous GCs in PD-1/PDL-1 Blockage

The role of endogenous GCs in the response to checkpoint blockade was analyzed by Acharya and collaborators who demonstrated that the deletion of the GR on CD8+ TILs or the inhibition of monocyte-macrophage steroidogenesis in TIME improves the control of tumor growth. According to these authors, the presence of an active signal of endogenous GCs can promote dysfunctional T cells and correlates with the failure of checkpoint blockade therapy in preclinical tumor models and in patients with melanoma [107]. Furthermore, endogenous GCs are increased during stressful conditions. Hyperactivity of the HPA axis, leading to an excessive release of endogenous GCs, is observed in liver cancer patients with depression and in a murine model of depression. In these studies, high levels of GCs led to the development of hepatocellular carcinoma via increased expression of PD-1 in tumor-infiltrating NK cells [108]. Furthermore, in a mouse model of depression established in tumor-bearing mice, lung cancer worsened in “depressed” mice due to changes in the TIME, including increased PD-L1 expression and reduced cytotoxicity of CD8+ T cells [109]. This again suggests a connection between neuroendocrine disorders, where GCs are increased, and immunity against tumors, where the PD-1/PD-L1 system is active.

In contrast, an advantageous effect of endogenous GC peak due to HPA axis activation in the early period of murine cytomegalovirus infection was demonstrated. Endogenous GCs induce selective upregulation of PD-1 on NK cells, which in turn limits the production of IFN-γ by splenic NK cells and prevents immunopathology without compromising viral clearance [110].

#### 4.2.3. GCs and Resistance to PD-1/PDL-1 Blockage

Although studies are still needed in this area, both exogenous and endogenous GCs could contribute to resistance to immune therapy against PD-1/PD-L1 in multiple ways. For effective targeting and inhibition of PD-1/PD-L1, with the goal of restoring an anti-tumor T cell response, a T cell must recognize a tumor antigen bound to major histocompatibility complex (MHC) proteins on an APC surface to become activated. If this activation is not well regulated, resistance to therapy can occur [111]. GCs may interfere in various ways with T cell activation. GCs may decrease the presentation ability of APCs, increase the number and function of Tregs, or directly inhibit T cell activation [67]. These effects may render immunotherapy ineffective or generate resistance to therapy (Figure 5).

##### Effect on Macrophage and DC

As mentioned above, resistance can be initiated by ineffective presentation of tumor antigens. GCs downregulate MHC molecules and induce tolerogenic macrophage and DC phenotypes. Some studies also suggest that GCs have the same effect on cancer cell MHCs [112]. An immunosuppressive microenvironment indirectly augments PD-1 expression and GCs through modulating lymphokine secretion and the expression of adhesion molecules [67,76], as well as directly regulating the expression of PD-1 on TILs [93].

##### Effect on Tregs

Furthermore, an increase in Tregs in the TIME can suppress not only host anti-cancer immunity but also diminishes the efficacy of immune therapy [113]. The role of the PD-1/PD-L1 pathway in the development and function of Tregs and that of Tregs on PD-1/PD-L1 function is conflicting and constantly evolving. Tregs in the TIME induce high levels of PD-1 on CD8+ T cells, which then become responsible, beyond a threshold limit, for anti-PD-1 resistance [114]. PD-L1, expressed on APCs and tumor cells, inhibit T cell responses by binding to PD-1, promoting the induction and maintenance of Tregs [115]. PD-1 signaling drives human Th1 cells to a Treg phenotype, impairing cell-mediated immunity [116]. This suggests a dual immunosuppressive role for the PD-1/PD-L1 pathway in T cells, via direct dampening of TCR signaling and induction of Tregs. However, mice that selectively lack PD-1 in Tregs exhibit an activated phenotype and have enhanced immunosuppressive function, which is responsible for improved experimental autoimmune encephalomyelitis and protection from diabetes in nonobese diabetic (NOD) mice [117]. Interestingly, this points to a new mechanism, mediated by Treg PD-1, for the control of immune tolerance [118].

Despite this protection during autoimmune disease, a high infiltration of Tregs is associated with poor survival in various types of cancer [119]. According to several studies, blocking the PD-1/PD-L1 interaction using nivolumab increases the proportion of Tregs in patients with oral cavity squamous cell carcinoma and this increase has been associated with a more favorable prognosis [120,121]. In patients with advanced non-small cell lung cancer, high levels of TGF-β and Tregs have become markers for predicting the response to PD-1/PD-L1-blocking therapy [122]. Immunosuppression resulting from an increase in Tregs plays a role in liver metastases, demonstrated by a mouse model of liver metastases, in which resistance to anti-PD-1 therapy was associated with increasing numbers of Tregs expressing CTLA-4. In this model, inactivating Tregs rescued the therapeutic function of PD-1 blockade [123]. Increasing Treg numbers and function, measured by inducible co-stimulatory molecule (ICOS) and proliferation marker Ki67 expression, contributes to immunosuppression and induces apoptosis of circulating CD4+ and CD8+ T cells [101]. In light of the contrasting role of Tregs during immunotherapy and in the induction of resistance, the effect of GCs could be either deleterious or beneficial. However, how GCs contribute to the development of resistance to PD-1/PD-L1 therapy involving Tregs remains unknown.

##### Effect on Activated Lymphocytes

Of particular interest is the opposing activity of PD-1/PD-L1-blocking therapy and GCs on the transcription pathways of activated lymphocytes. PD-1-blocking drugs restore the activity of pathways inhibited by PD-1, including PI3K/Akt and PLCγ/Ras/MEK/ERK. GCs inhibit those same pathways, counteracting the effects of PD-1-blocking drugs by generating opposite outcomes on tumor-specific activated T cell proliferation, survival, apoptosis, and cell cycle progression.

When these pathways that are essential for growth, proliferation, and spread of tumor cells are activated by genetic mutations, they may lead to increased PD-L1 expression in cancer cells [23]. In addition to PD-1, PD-L1 expression is also important in inducing the inhibitory response. PD-L1’s upregulation on tumor cells creates favorable conditions for tumor progression. Genetic deletion of PD-L1 in cancer cells reduced PD-1 activation, leading to decreased tumor growth [124]. As previously mentioned, PD-L1 is also upregulated in cancer cells by pro-inflammatory cytokines, such as IFNγ, TNF-α, and IL-6 [125,126,127]. Surprisingly, recent data suggest that treating tumor cell lines with GC promotes anti-tumor activity by decreasing PD-L1 expression [128]. In contrast, GCs induce PD-L1 expression in human DCs through transcription of a GR target gene, GC-induced leucine zipper (GILZ). GILZ is upregulated by DEX treatment in human DCs and induces a tolerant DC phenotype, increasing the expression of PD-L1. In vivo studies using healthy subjects demonstrate that DEX treatment increases PD-L1 expression in human DCs [129]. Accordingly, upregulation of GILZ via DEX increases the expression of PD-L1 in human DCs [130] and is required for in vitro PD-L1 expression [131]. Silencing GILZ leads to a decrease in PD-L1 expression associated with IL-12 secretion and T cell activation [132]. Furthermore, in a murine model of stress, high cortisol plasma levels impaired IFNγ-induced T cell activation via upregulation of GILZ in DCs. In addition, in this study, GILZ depletion in DCs improved therapeutic response to PD-1 blockade [133] (Figure 5). However, there is no current evidence for GR’s transcriptional regulation of PD-L1 in DCs. Conversely, in vitro studies, in contrast to in vivo observations [129], demonstrate that GCs do not alter PD-L1 expression in DC cells [134,135]. The activity of GCs on DC is, therefore, immunosuppressive, both directly and through GILZ-induced PD-L1 expression, thus favoring tumor immune escape. Although PD-L1 expression is used as a predictive marker for the response to blocking therapy [136], the variable effects of GCs on the PD-1/PD-L1 pathway may produce different outcomes, depending on the type of tumor that is targeted by therapy.

## 5. Conclusions: Does the Immune System Become Confused?

The genes that regulate PD-1/PD-L1 expression and the signaling pathways activated by PD-1 are often the same ones regulated by GCs. Therapeutically blocking the PD-1/PD-L1 pathway often coexists with endogenous stress-induced GCs and, when necessary, with exogenous administration of therapeutic GCs. Therefore, different and often conflicting signals communicate with the immune system and GCs may counteract the effect of PD-L1-blocking treatment on immune system activity. The complexity of the interactions is, on one hand, due to the many effects of GCs and, on the other hand, due to the effects of blocking therapy involving not only CD8+ T cells but also other cell types involved in a “non-canonical block”, as recently suggested [137]. The use of GCs is often necessary to counteract the adverse effects of immunotherapy [103,138], and the indications for using GCs are cerebral metastases, diarrhea, or fatigue [139].

However, GCs may increase or decrease cancer growth and spread by direct activity on the tumor, the immune system, or the TIME [70]. Yang discussed the GC double-edged sword in cancer and COVID therapy, commenting that their effects depend on the dose, the duration of therapy, and, when combined with PD-1-blocking therapy, the time of administration of both drugs [140]. Although GC short-term treatment may not affect the outcome of PD-1 inhibitor therapy, there are no studies that examine the combination of long-term GC treatment and PD-1 inhibitors compared with PD-1 inhibitors alone, in relation to patient outcomes.

Identifying the interactions between GCs and immunosuppressor drugs is crucial for the optimal use of immunotherapy. The confusion of the immune system mirrors our own confusion, as we have yet to dissect the plethora of signals generated by the two, often both necessary, types of therapy.

## Figures and Tables

**Figure 1 cells-10-02333-f001:**
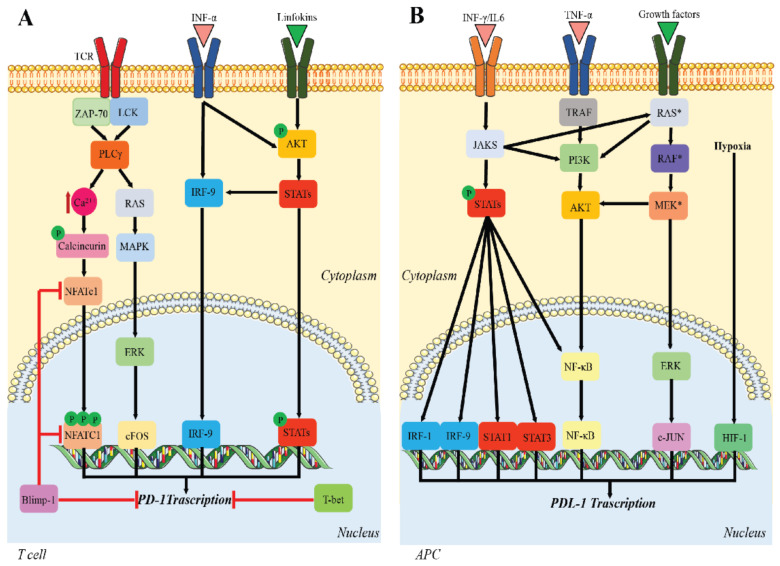
Regulation of PD-1/PD-L1 expression. (**A**) Upstream signaling pathways leading to *PD-1* transcription; (**B**) signaling pathways involved in *PD-L1* transcription. Multiple pathways promote PD-L1 expression. Red = inhibitory signaling; black = activating signaling.

**Figure 2 cells-10-02333-f002:**
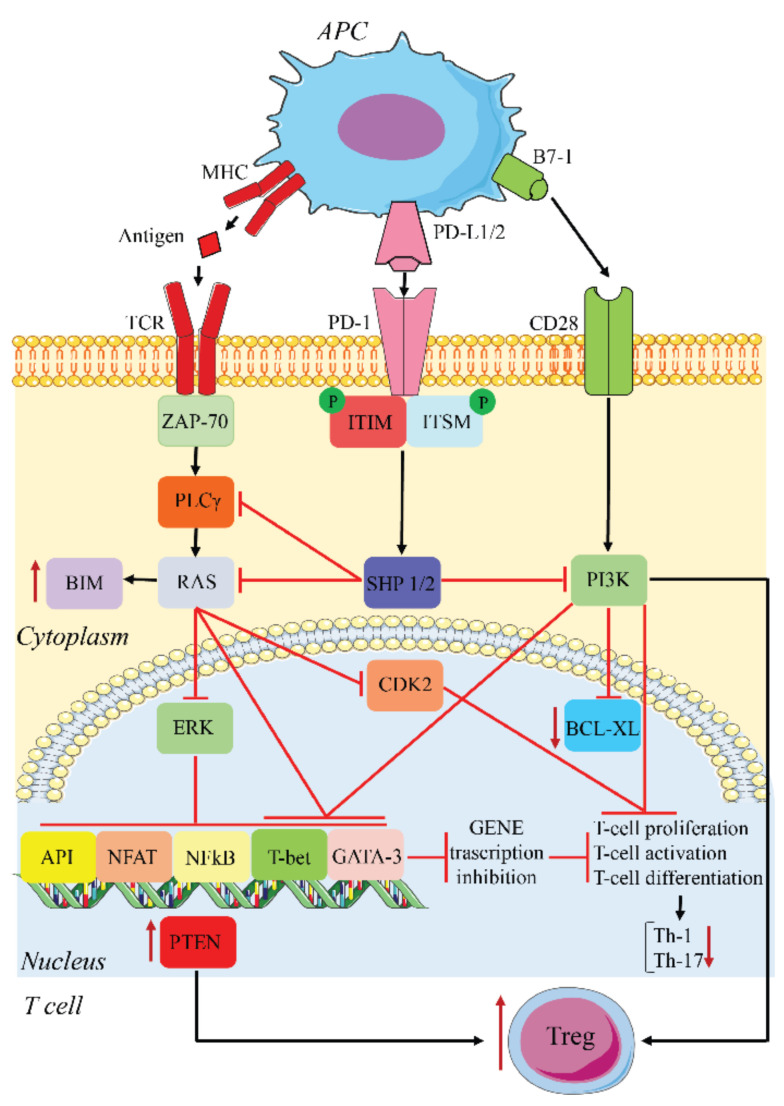
PD-1/PD-L1 inhibitory pathway. APCs present antigens released from tumor cells to T cells, leading to T cell activation and PD-1 expression. PD-1 binds PD1-L1/2 expressed on the surface of both APCs and tumor cells, leading to inhibition of TCR and CD28 signaling. This results in an inhibition of cell proliferation, activation, and differentiation. Th-1 and Th-17 differentiation is inhibited, while Tregs are increased. The upregulation of PTEN participates in the latter.

**Figure 3 cells-10-02333-f003:**
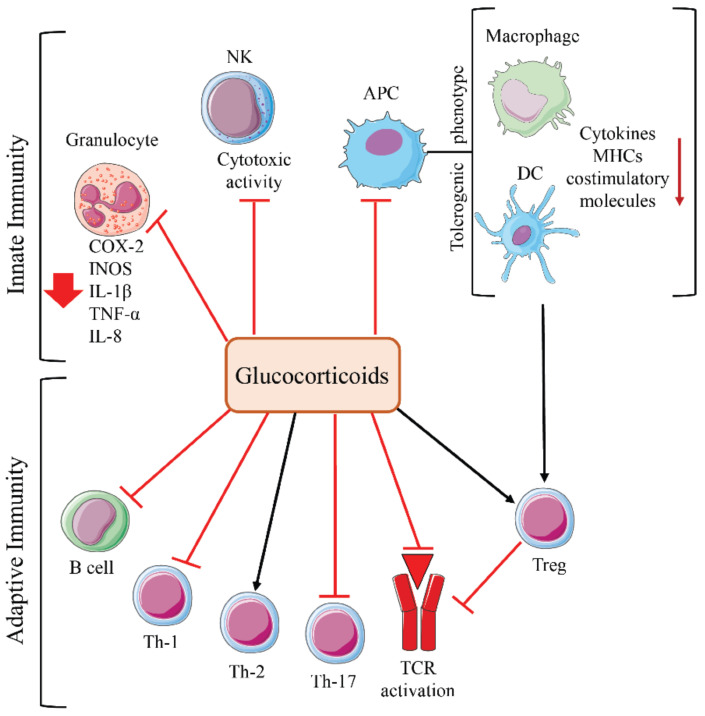
Effects of glucocorticoids on innate and adaptive immunity. Cells involved in GC-mediated immunosuppression. Red = inhibitory signaling; black = activating signaling.

**Figure 4 cells-10-02333-f004:**
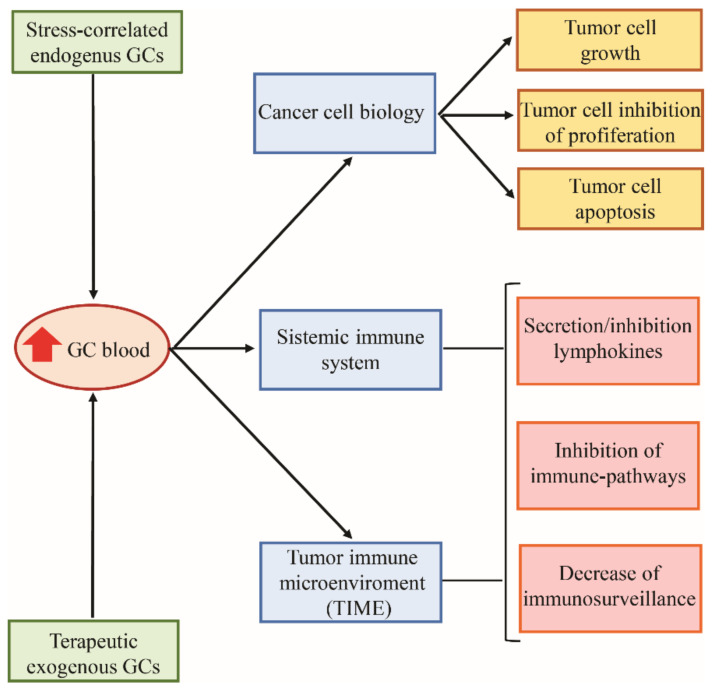
The effects of glucocorticoids on tumor cells.

**Figure 5 cells-10-02333-f005:**
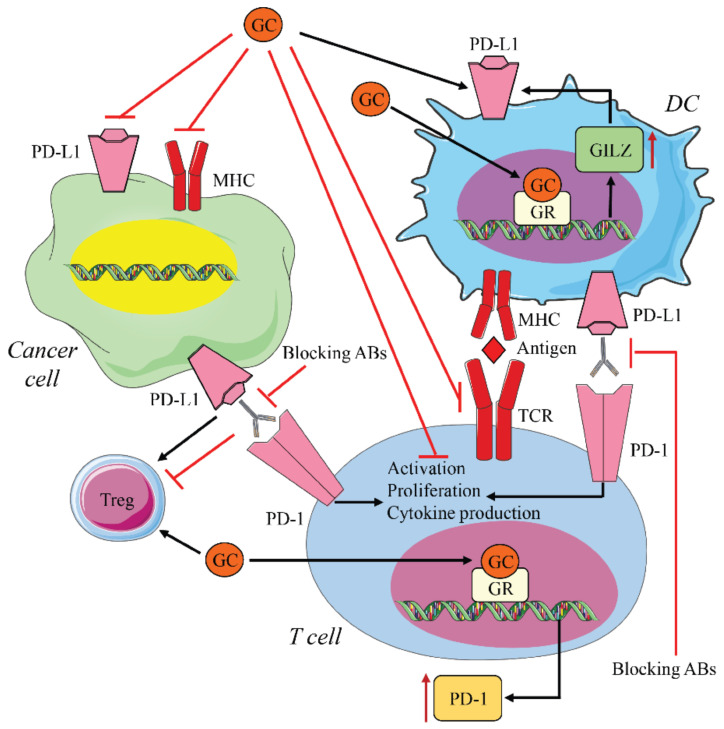
Interaction between glucocorticoid and PD-1/PDL-1 signals. Blocking antibodies (anti-PD-1/PD-L1) induce T cell reactivation by restoring T cell proliferation and cytokine production. GCs counteract this effect by inhibiting T cell activation through several mechanisms. GC binding to the GR induces *PD-1* transcription in T cells and NK cells, hindering the efficacy of blocking therapy. GCs also upregulate PD1-L in DC cells, through GILZ expression, decreasing the efficacy of immunotherapy. In contrast, GCs inhibit PD-L1 on tumor cells, thus exerting anti-tumor activity GC and anti-PD-1/PDL-1 immunotherapy have different effects on Tregs. In some cases, they both increase Tregs; in other cases, GCs increase Tregs while immunotherapy decreases them.

## Data Availability

Not applicable.

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
