# Peer review of "Glucocorticoid and PD-1 Cross-Talk: Does the Immune System Become Confused?"

_cells, 2021, doi:10.3390/cells10092333_

Round 1

Reviewer 1 Report

It is well established that glucocorticosterids have a significant impact on the immune system. They are frequently administrated to cancer patients to reduce the side effect of immunotherapy. In the paper the Authors thoroughly discuss the effect of GCs on the immunotherapy focusing on PD-1/PD-L1 pathway underling potential counterproductive effects of GCs and blocking mAbs combination therapies in patients with cancer.  It is worth mentioning that in this place the Gaucher et al. analysed the effect of GCs on immunotherapy and concluded that: “Systemic corticosteroid use for an irAE does not impact OS or the tumor response, whereas use for other indications (themselves often associated with a worse prognosis) does. Topical and inhaled steroids do not have a negative impact on OS.” [Ther Adv Med Oncol. 2021 Feb 27;13:1758835921996656. doi: 10.1177/1758835921996656. eCollection 2021.]

I have some comments and suggestions which the Authors may want to address:

Figure 1.

Why the IRF1 has not  been included in Fig 1. B?  

The quality of the figure should be improved.

Line 247

Please double check GATA3 or GATA1

Figure 3.

Please pay attention to this figure. In the line 259 it is stated that GC inhibited Th-17 cells but according to Fig.3 GCs promoting Th17 cells; also Th2 cell subpopulation are promoted not inhibited.

Tumor immune microenviroment. In my opinion the TIME abbreviation is more frequently use than TIM.

Author Response

The manuscript has been revised based on all Reviewer concerns. We would like to thank the Reviewers for their insightful comments that have improved the quality of this paper. All substantial changes are reported in red.

Reviewer # 1

- The suggested paper (Gaucher et al., 2021, Ther Adv Med Oncol) was added and briefly discussed.

- In the new Figure 1B, IRF1 has been included.

- We agree with the Reviewer; the quality of the figures was poor. For this reason, we have increased the resolution and the size of all figures.

- Line 247: both GATA-3 and GATA-1 are modulated by glucocorticoids. GATA-1, as suggested by the Reviewer, has now been added in the text.

- I thank the Reviewer for focusing my attention on Figure 3, which was wrong. Corrected, new Figure 3 shows that glucocorticoids inhibit Th-1 and Th-17 and promote Th-2 differentiation. Furthermore, glucocorticoids inhibit B cell function.

- As suggested, the abbreviation “TIM” has been replaced by “TIME”.

Reviewer 2 Report

In the manuscript, Adorisio et al. reviewed the current knowledge regarding the crosstalk between PD-1/PD-L1 and GC biology. Overall, this is a comprehensive review that covers a wide range of literature. I think this is suitable for publication in Cells. A few specific comments are following.

Specific comments:

- This review mainly describes the outcome of pharmacological concentration of GC. Although this review focuses on therapeutic application of GC in the PD-1/PD-L1 inhibitor therapy, it would be beneficial if authors address a role of GC in its physiological concentration. Given the pleotropic effect of GC depending on its dosage, it would make more sense to discuss the physiological condition as well as supraphysiological condition.

- The 4.2 section (Cross-talk between GC and PD-1 pathways in tumor immune evasion) is very long and describes multiple aspects of GC and PD-1 pathways. It is easy to get lost in so much text, particularly in the case of paragraphs that continue one after another, which requires better organization. One suggestion I have for the authors is splitting and subcategorizing the section, for example, by cell types. Or the authors could make a table summarizing the interaction to navigate readers through it.

- To rule out the effect of GC in T cells during PD-1/PD-L1 inhibitor therapy, analyzing T cell specific GR KO mice might be informative. As such, the authors could cite a paper by Acharya et al. (Immunity, 2020) and mention how genetic mouse models could help understand crosstalk between GC and immune checkpoint blockade. Likewise, other GR KO mice in different Cre background would be useful.

- The resolution of figures needs to be improved, especially for Figure 1, 2, and 3.

- Typos: (1, 3). Page 1, line 38; (27, 28). Page 3, line 97.

- Missing abbreviation: PDCD1 programmed cell death protein 1. Page 2, line 87. This can be shown in Page 1, line 7.

- Typo: Space. Page 6, line 256.

- Typos: TMI should be corrected to TIM. Page 10, line 385; Page 11, lines 427 and 430; Page 13, line 505.

Author Response

The manuscript has been revised based on all Reviewer concerns. We would like to thank the Reviewers for their insightful comments that have improved the quality of this paper. All substantial changes are reported in red.

Reviewer # 2

- As suggested by the Reviewer, we briefly discussed the role of endogenous glucocorticoids on the modulation of PD-1/PDL-1 (references 107-110).

 - Section 4.2. (Cross-talk between GC and PD-1 pathways in tumor immune evasion) has been divided into paragraphs, also based, as suggested, on the cell type.

- As suggested, the paper by Acharya and al. (Immunity. 2020; 53 (3): 658-71) was cited as an example of mouse genetic modification that can help understand the cross-talking between PD-1 and GCs.

- The resolution of the figures has been improved.

- All typos have been fixed.

- Programmed cell death 1 (PDCD-1).

- The “tumor immune microenvironment” was abbreviated as TIME at the suggestion of the first Reviewer.